# Hemophagocytic Lymphohistiocytosis Associated with Immunological Checkpoint Inhibitors: A Pharmacovigilance Study

**DOI:** 10.3390/jcm12051985

**Published:** 2023-03-02

**Authors:** Laurine Diaz, Benjamin Jauzelon, Anne-Charlotte Dillies, Cosette Le Souder, Jean-Luc Faillie, Alexandre Thibault Jacques Maria, Pascale Palassin

**Affiliations:** 1Department of Medical Pharmacology and Toxicology, CHU Montpellier, Montpellier University, 34000 Montpellier, France; 2Internal Medicine & Immuno-Oncology (MedI2O), CHU Montpellier, 34000 Montpellier, France; 3Desbrest Institute of Epidemiology and Public Health (IDESP), Montpellier University, INSERM, 34000 Montpellier, France; 4Institute for Regenerative Medicine and Biotherapy (IRMB), Montpellier University, 34000 Montpellier, France

**Keywords:** hemophagocytic lymphohistiocytosis, immune checkpoint inhibitors, pharmacovigilance, auto-immune disorders, adverse drug reactions

## Abstract

Background: Acquired hemophagocytic lymphohistiocytosis (HLH) is a rare but potentially fatal condition characterized by hyperactivation of macrophages and cytotoxic lymphocytes, combining a series of non-specific clinical symptoms and laboratory disorders. Etiologies are multiple: infectious (mainly viral) but also oncologic, autoimmune or drug-induced. Immune checkpoint inhibitors (ICI) are recent anti-tumor agents associated with a novel profile of adverse events triggered by immune system over-activation. Here, we sought to provide a comprehensive description and analysis of HLH cases reported with ICI since 2014. Methods: Disproportionality analyses were performed in order to further explore the association between ICI therapy and HLH. We selected 190 cases, 177 from the World Health Organization pharmacovigilance database and 13 from the literature. Detailed clinical characteristics were retrieved from the literature and from the French pharmacovigilance database. Results: The cases of HLH reported with ICI concerned men in 65% of cases with a median age of 64 years. HLH occurred in an average of 102 days after the initiation of ICI treatment and mostly concerned nivolumab, pembrolizumab and nivolumab/ipilimumab combination. All cases were considered serious. Most cases presented a favorable outcome (58.4%); however, death was reported for 15.3% of patients. Disproportionality analyses showed that HLH was seven times more frequently reported with ICI therapy than with other drugs and three times more than with other antineoplastic agents. Conclusions: Clinicians should be aware of the potential risk of ICI-related HLH to improve the early diagnosis of this rare immune-related adverse event.

## 1. Introduction

Immune checkpoint inhibitors (ICI), by targeting the cytotoxic T-lymphocyte-associated protein 4 (anti-CTLA-4) or the Programmed Death receptor 1 (anti-PD-1) or its ligand (anti-PD-L1), can cause immune-related adverse events (irAEs) that may affect all organs, in line with their mechanism of action, i.e., restoring immune response against tumor cells. The most frequently reported adverse events are dermatological toxicities in more than 30% of exposed patients, followed by digestive (approximately 15%), endocrine (10%) and hepatic (5 to 10%) toxicities [1]. In most cases, ICI-induced adverse events occur within 3 to 6 months, although they may occur later, up to one year after initiation [2,3].

Post-marketing surveillance of drugs through pharmacovigilance systems is particularly useful to detect adverse events in real life, especially rare adverse events that were not identified in clinical trials.

Reactive hemophagocytic lymphohistiocytosis (rHLH) is a severe condition resulting from immune activation and dysregulation. It is characterized by hyperactive macrophages and cytotoxic lymphocytes leading to hypersecretion of pro-inflammatory cytokines and extreme inflammation possibly associated with rapid tissue destruction, multiple organ failure and death [4,5]. HLH is often under-diagnosed due to the non-specificity of clinical symptoms and abnormal laboratory results. Some other inflammation diseases can present similar manifestations, notably the immune reconstitution inflammatory syndrome (IRIS) reported in HIV-infected individuals after antiretroviral treatment initiation. It can induce inflammatory reactions in tissues related to the presence of opportunistic bacterial infections or to visceral Kaposi sarcoma, Kaposi sarcoma-associated herpesvirus inflammatory cytokine syndrome [6]. Therefore, the Saint-Antoine diagnosis composite score (or HScore) has been proposed combining clinical and laboratory data and predicting the probability of HLH [7]. The etiologies of reactive HLH (rHLH) are multiple. Infection is the most frequent cause with mainly viral origin (EBV, herpes viruses and cytomegalovirus) but bacterial, parasitic or fungal etiologies are also reported. Other etiologies include autoimmune diseases (such as systemic lupus and Still’s disease) and malignancies, mainly hematological. Drug-induced HLH have been reported with drugs generally associated with a hypersensitivity syndrome (DRESS) such as lamotrigine or sulfamethoxazole and with immunotherapies for cancer such as immune checkpoint inhibitors (ICI) or chimeric antigen receptor CAR-T cell therapy) [8,9]. Indeed, patients with severe Cytokine Release Syndrome, the most frequent adverse reaction with CAR-T cells, present similar clinical and laboratory features to those of rHLH [10]. In the past four years, cases of ICI-induced HLH have been reported in the literature [11,12,13,14,15,16,17,18,19,20,21,22,23,24,25,26,27,28,29,30,31] with little information about risk factors, clinical presentation and outcomes.

The objective of this study is to present a comprehensive review and analysis of all cases of HLH reported with ICI both through pharmacovigilance databases and the literature in order to better characterize this under-recognized irAE. 

## 2. Materials and Methods

We first searched the World Health Organization (WHO) international pharmacovigilance database, VigiBase^®^, for all individual case safety reports (ICSR) using the Preferred Term “Hemophagocytic lymphohistiocytosis” of the Medical Dictionary of Regulatory Activities (MedDRA 25.0), associated with at least one ICI (anti-CTLA-4, anti-PD1 or anti-PD-L1), from October 2014 to July 2022. VigiBase^®^ is a global database collecting more than 33 million adverse drug reaction reports from over 150 countries worldwide. ICSR provide data about patient’s age, sex, country of origin, date of report, reporter qualification, drug, adverse effects and seriousness criteria according to the International Council for Harmonization of Technical Requirements for Pharmaceuticals for Human Use (ICH). Time to onset and time to resolution can be calculated if the start date and the end date of the ICI treatment and of the HLH reaction are precisely mentioned. Since Vigibase^®^ ICSRs do not contain detailed medical history, we subsequently searched for all cases of HLH associated with ICI in the French pharmacovigilance database which presents more detailed clinical descriptions and allows the analysis of the Saint-Antoine score criteria.

Pharmacovigilance signal detection was conducted with disproportionality analyses using the case/non-case method. Disproportionate reporting, meaning higher-than-expected number of adverse reaction reports compared with other reactions recorded in the database, was analyzed by calculating the Reporting Odds Ratios (ROR) [32]. Cases were all HLH reports and non-cases were all other serious drug-related adverse reactions recorded in Vigibase^®^. ICI drug exposure was compared between cases and non-cases to all other drugs, and to all other anti-neoplastic drugs in a second analysis. A disproportionate reporting signal was considered if the lower limit of the 95% confidence interval (95% CI) of the ROR exceeded 1.

Finally, a review of the cases reported in the literature was performed in the Medline database using the terms “Hemophagocytic lymphohistiocytosis” and “Immune Checkpoint Inhibitor”.

## 3. Results

Among the 3106 cases of HLH reported in VigiBase^®^, 177 (5.7%) were associated with an ICI agent. Among these cases, 30 originated from the French pharmacovigilance database. The literature review identified 21 relevant articles and 13 additional cases of ICI-induced HLH that were not found in Vigibase^®^ after a search for duplicates. A total of 190 cases were included in the analysis and are presented in Table 1.

Reporters of ICI-induced HLH were primarily health care professionals. ICI was most often the only suspected drug (in 75.3%, n = 143 cases). In 64% of cases, HLH occurred under ICI monotherapy. The most frequently reported ICIs were nivolumab (21.3% in monotherapy and 33.8% in combination with ipilimumab) and pembrolizumab (30.2% of the cases). The most frequent co-reported drugs were sulfamethoxazole-trimethoprim, trametinib (anti-MEK) and dabrafenib (anti-BRAF), in 6.3% (n = 12), 4.7% (n = 9) and 3.2% (n = 6) of the cases, respectively [33].

The mean time to onset (calculable for 81 cases) was 106 days with a median of 50 days. Regarding HLH outcome, 32.6% of patients recovered and 24.2% were recovering at the time of reporting. The mean time to resolution, when calculable (n = 13) (reaction resolved at the time of the report), was 47 days. All cases were considered serious and a fatal outcome was reported for 15.3% of patients.

Disproportionality analyses showed that, among all serious adverse drug reactions reported in VigiBase^®^, HLH was nearly seven times more frequently reported with ICI than with all other drugs (ROR = 6.58; 95% CI: 5.65–7.66) and was also nearly three times more reported with ICI than with all other antineoplastic drugs (ROR = 2.87; 95% CI: 2.44–3.37), including CAR-T cells drugs.

Detailed clinical data extracted from the literature cases (n = 13, Appendix A) and the French pharmacovigilance database (n = 30) are presented in Table 2. The most frequent diagnosis criteria reported were hyperferritinemia (83.7%), fever (81.4%) thrombocytopenia (76.7%) and anemia (69.8%). Corrective treatment was most often corticosteroids alone, sometimes combined with etoposide. Tocilizumab was used alone in three cases. More than 80% of these cases presented a favorable outcome.

## 4. Discussion

This study presents the largest series of HLH cases reported with ICI therapy. Disproportionality analyses clearly identified a pharmacovigilance signal for HLH associated with ICI when compared to other antineoplastic drugs including targeted therapy and CAR-T cells drugs. Hence, ICI treatment is, among oncological therapies, one of the major etiologies of reactive HLH.

Comparing our results to other cases of rHLH related to classic etiologies, such as infection, auto-immune disease or neoplasm, ICI-induced HLH showed similar mean age of onset (around 50 years old), but affected men more than women (79% vs. 60%). This may be explained by a higher rate of autoimmune diseases associated with rHLH in other studies and with the higher cancer incidence in men [4,7]. Regarding the clinical symptoms of the patients, we report fewer splenomegaly and hepatomegaly cases (respectively, 28% and 9% vs. 69% and 67%). Another interesting difference is the low rate of haemophagocytosis on bone marrow analysis (46.5% vs. 85%), which is a major criterion for HLH in the HScore, and can compromise differential diagnosis in our study, but also the lack of clinical data in global pharmacovigilance reports. Other laboratory results are quite similar, but data regarding the precise level of ferritinemia is lacking. One way to improve HLH diagnosis may be to analyze glycosylated ferritin and CD25 blood rate [34]. It is shown that elevation of soluble CD25 is a valid diagnosis criterion [35]. However, it is not generally available in clinical practice.

Of note, only 13% of the patients in our series presented a known underlying immunodepression, excluding cancer, which is quite a low rate as it is known to be a frequent predisposing factor (approximately 45% [7]).

Among malignancy-related HLH (mHLH), it is now well established that hematological diseases, and particularly T-cell lymphoma and B-cell lymphoma, are more common causes than solid cancers [36]. These hematological diseases have been rarely reported in our series. This can be explained by the less developed use of ICI therapy in hematologic cancers than in solid tumors. However, considering the very rapid expansion of ICI indications, special attention should be paid to the early clinical or laboratory signs of HLH.

Concerning HLH management, we noticed that a majority of patients (53%) required only corticosteroids. In a study that focused on HLH associated with EBV positive lymphoma, only one patient (2.5%) received corticosteroids alone, with a wide use of poly-chemotherapy [37]. The same rate is reported in another study about mHLH [38]. However, in HLH secondary to autoimmune disease, the use of high dose of corticosteroids in monotherapy is more frequent (approximately 50%) [39], which is consistent with the mechanism of immune mediated toxicity of ICI. Usually, patients with rHLH refractory to steroids are treated with immunosuppressive drugs such as cyclophosphamide [39], or using etoposide, a drug that may even be used in front line in this situation [40].

Again, a more precise analysis of the treatment modalities (i.e., dose of steroids and/or type of immunosuppressive drugs) is lacking in this peculiar situation, while specific protocols such as HLH-94 and 2004 treatment regimens are proposed in other situations, but have not been evaluated in ICI-related rHLH [41].

Interestingly, regarding ICI-related rHLH outcome, we observed a lower mortality rate (approximately 15%) compared with other rHLH (20–40%) [4,39].

The present work questions the possible association between ICI mainly involving adaptive T cell immunity and innate immunity. Innate immunity is essential for building and maintaining adaptive immunity and fully integrates the cancer-immunity cycle. The role of IFN gamma released in the tumor microenvironment appears to be a key regulator of this interaction [42,43]. Detection of tumor cells induces activation of innate immune cells such as NK cells, promoting tumor cells death. Thereby, it generates new antigenic signals through antigen presenting cells infiltrating tumor microenvironment which amplifies the immune response. It has been recently showed that tumor-associated macrophages increase over time in murine and human cancer and their PD-1 expression inversely correlates with their phagocytic potency against tumor cells [44]. The PD-1/PD-L1 blockade seems to increase macrophage tumor cell phagocytosis. Beyond their direct anti-tumor effect, innate immune cells activated by tumor cells participate in all stages of the production and activation of T lymphocytes directed against cancer cells [45]. Therefore, it might be hypothesized that an association exists between deregulated immune activation in HLH and immune activation toxicities seen with ICI through a complex interplay of innate and adaptive immune cells.

Our study presents some limitations notably inherent to pharmacovigilance data. Indeed, the reporting rate cannot be interpreted as real incidence, notably due to the underreporting of adverse drug reactions. In our series, the low rate of cases both described in the literature and reported in VigiBase further illustrates this under-reporting phenomenon and the scarcity of HLH in general. Disproportionality analyses allowed us to evaluate the strength of the association between HLH and ICI but based on reported data only, this method cannot assert causality, even if it has been demonstrated to be useful to identify drug-related risks [46].

Another limitation of VigiBase^®^ analyses, as mentioned above, is missing data, such as past medical history, complete work-up, co-reported affections and management strategies. In this context, it may strongly impact HLH diagnosis, notably through the precise evaluation of the HScore and potential confounders or differential diagnoses. For instance, we did not precisely know the number of patients with concomitant sepsis, which is a main cause of HLH by itself. In addition, we only retrieved a few cases with EBV reactivation. It could be interesting to better identify the patients with viral infection or reactivation, such as EBV reactivation, which is known to be a primary cause of HLH, especially in patients with malignancy disease (co-factor in 24% of those patients and up to 88% in patients under chemotherapy) [47]. Of note, patients with EBV reactivation can benefit from rituximab treatment [48]. Conflicting data are reported regarding the role of EBV and ICI: some studies suggest EBV infection or reactivation under ICI with various clinical presentations (cerebellar ataxia [49], colitis [50], etc.), but paradoxically, it seems that EBV-related HLH may also benefit from a treatment with nivolumab [51].

Despite all these limitations, the combination of the largest data set using post-authorization surveillance data together with the precision of a qualitative sample from the French pharmacovigilance reports brings relevant original data regarding the risk of HLH under ICI.

## 5. Conclusions

ICI therapy may trigger HLH, a rare but serious immune-related adverse event that may be under-diagnosed in this situation. We describe the largest series of ICI-induced HLH and showe an increased reporting compared with other antineoplastic agent. However, data are still lacking to better apprehend this complication in the context of ICI use and define the most appropriate treatment.

## Figures and Tables

**Table 1 jcm-12-01985-t001:** Characteristics of ICI-induced HLH cases reported in VigiBase^®^ (n = 177) and in the literature (n = 13).

Characteristics	Cases (n = 190)
**Age** (years), median (min-max) (n = 161)	64.0 (2–101)
**Gender**, n (%) (n = 182)	
Male	118 (64.8)
Female	64 (35.2)
**Reporter type**, n (%)	
Health professional	186 (97.9)
Other	4 (2.1)
**Reporting year**, n (%)	
2022	17 (8.9)
2021	37 (19.5)
2020	33(17.3)
2019	53 (27.9)
2018	28 (14.7)
2017	12 (6.3)
2016	6 (3.2)
2015	3 (1.6)
2014	1 (0.5)
**Exposure to ICIs**	
Monotherapy, n (%)	123 (64.0)
Anti PD-1	101 (82.1)
Nivolumab	41 (40.6)
Pembrolizumab	58 (57.4)
Cemiplimab	2 (2.0)
Anti PD-L1	15 (12.2)
Atezolizumab	15 (100.0)
Durvalumab	0 (0)
Anti CTLA-4	7 (5.7)
Ipilimumab	7 (100.0)
Combination	69 (36.0)
Nivolumab/Ipilimumab	65 (94.2)
Pembrolizumab/Ipilimumab	2 (2.9)
Nivolumab/Ipilimumab/pembrolizumab	2 (2.9)
**Indication**, n (%)	
Melanoma	76 (40.0)
Lung cancer	42 (22.1)
Leukemia	7 (3.7)
Renal cancer	10 (5.3)
Bladder cancer	5 (2.6)
Hodgkin’s disease	2 (1.0)
Others	29 (15.3)
Unknown	19
**Mean time to onset** (days) (n = 81)	102
Median	50
**Evolution** n (%)	
Recovered	62 (32.6)
Recovered with sequelae	3 (1.6)
Recovering	46 (24.2)
Not recovered	18 (9.5)
Death	29 (15.3)
Unknown	32
**Mean time to recovering** (days) (n = 13)	47.3
Median	20.0

**Table 2 jcm-12-01985-t002:** Clinical description of ICI-induced HLH reported in the French pharmacovigilance database (n = 30) and the literature (n = 13).

Characteristics	HLH (n = 43)
**Age** (years), median (min-max)	50 (2–76)
**Gender, n (%)**	
Male	34 (79)
Female	9 (21)
**Saint Antoine score criteria, n (%)**	
Known underlying immunosuppression	4 (13)
Fever	35 (81.4)
Hepatomegaly	4 (9.3)
Splenomegaly	12 (27.9)
Anemia	30 (69.8)
Leucopenia	21 (48.8)
Thrombopenia	33 (76.7)
Hyperferritinemia	36 (83.7)
Hypertriglyceridemia	24 (55.8)
Hypofibrinogenemia	14 (32.6)
Cytolysis	27 (62.8)
Hemophagocytosis	20 (46.5)
**Support, n (%)**	
Steroids	23 (53.5)
Etoposide and steroids	6 (13.9)
Mycophenolate mofetil and steroids	1 (2.3)
Etoposide, tocilizumab and steroids	2 (4.6)
Tocilizumab	3 (7.0)
Unknown	8
**Evolution, n (%)**	
Recovered	19 (44.2)
Recovering	16 (37.2)
Not recovered	3 (7.0)
Death	3 (7.0)
Unknown	2
**Risk factor for HLH present**	
Ongoing infection	14
Including EBV Reactivation	6

## Data Availability

The data were provided by the Uppsala Monitoring Centre and the French Agence Nationale de Sécurité des Médicaments et des produits de santé (ANSM). These databases are accessible by French regional pharmacovigilance centers.

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
