# Peer review of "Hemophagocytic Lymphohistiocytosis Associated with Immunological Checkpoint Inhibitors: A Pharmacovigilance Study"

_jcm, 2023, doi:10.3390/jcm12051985_

Round 1
Reviewer 1 Report
I like the paper. It is very thorough. I do think it would benefit from language editing
Author Response
We sincerely thank Reviewer#1 for his/her positive and encouraging comments. We made an effort to improve English language and style by reformulating some sentences and ideas. We also tried to clarify the peculiar methodology of disproportionality analyses which could be difficult to explain and understand. All these modifications appear with tracked changes in the revised version of the manuscript.
Reviewer 2 Report
I have read with interest this interesting PV study regarding secondary HLH due to cancer immunotherapy. I have the following comments to the authors in other to improve the paper.
1. line 15 in abstract" associating a set of" doesn't make sense, please rewrite; Possibly associated with ...
2. line 16 " biologic symptoms" - what is it? I do not think such a term exists
3. anti tumoral should ani-cancer or anti-tumor
4. biological data should be laboratory data
5. Introduction should also describe conditions that can manifest similarly- for example KCIS " Dumic I, Radovanovic M, Igandan O, Savic I, Nordstrom CW, Jevtic D, Subramanian A, Ramanan P. A Fatal Case of Kaposi Sarcoma Immune Reconstitution Syndrome (KS-IRIS) Complicated by Kaposi Sarcoma Inflammatory Cytokine Syndrome (KICS) or Multicentric Castleman Disease (MCD): A Case Report and Review. Am J Case Rep. 2020 Dec 3;21:e926433. doi: 10.12659/AJCR.926433. PMID: 33268763; PMCID: PMC7722771.
6. Another condition that also can have similar manifestations is CRS " Frey N, Porter D. Cytokine Release Syndrome with Chimeric Antigen Receptor T Cell Therapy. Biol Blood Marrow Transplant. 2019 Apr;25(4):e123-e127. doi: 10.1016/j.bbmt.2018.12.756. Epub 2018 Dec 23. PMID: 30586620."
7. HLH is a complex syndrome to diagnose so I am wondering who are " the others" who reported HLH in 4 cases ( Table 1)? Can we trust that it was indeed HLH?
8. Line 138, 139- splenomegaly and hepatomegaly are very subjective terms in the absence of imaging. Did all patients have abdominal US or CT scan?
9. What is the % of patients who had bone marrow biopsy?
Author Response
We sincerely thank both reviewers for their positive and encouraging comments. We made an effort to improve English language and style by reformulating some sentences and ideas. We also tried to clarify the peculiar methodology of disproportionality analyses which could be difficult to explain and understand. All these modifications appear with tracked changes in the revised version of the manuscript. We thank Reviewer#2 for his/her suggestions that we tried to apply accordingly.
- Line 15 in the abstract, we modified as rightly suggested the sentence about the description of HLH disease.
- Line 16 and other occurrences in the text, we modified the wrong terminology of ‘biological symptoms’ for ‘laboratory disorders, results or data’.
- We modified as suggested anti-tumoral for anti-tumor.
- As already mentioned, we modified all occurrences of ‘biological data’ for ‘laboratory results or data’.
- and 6. As suggested, we added in the introduction the relevant references about quite similar manifestation that can occur in HIV-infected patients with Kaposi-sarcoma and as a severe adverse drug reaction after CAR-T cells infusion.
- Since HLH is a complex syndrome to diagnose, the Reviewer#2 is wondering who are "the others" who reported HLH in 4 cases ( Table 1)? Can we trust that it was indeed HLH?
For these four cases from the United States, reporters are indeed not health professionals. However, it seems unlikely that the specific diagnosis of HLH was not secondary to medical assessment.
- Line 138, 139 - splenomegaly and hepatomegaly are very subjective terms in the absence of imaging. Did all patients have abdominal US or CT scan?
When the information on hepatomegaly, splenomegaly or hepatosplenomegaly was available (n=22 cases), a US or CT scan have been systematically performed.
- What is the % of patients who had bone marrow biopsy?
For all patients in whom haemophagocytosis have been reported, a myelogram has been performed. We found in all the French pharmacovigilance cases (n=30) the mention of 23 myelograms and one precision of bone marrow biopsy, which in total represent 80% of these cases. In the cases extracted from literature (n=15), we found 13 bone marrow biopsies (86.7%).
Round 2
Reviewer 2 Report
I would like to thank the authors for the detailed revisions of the paper, and for their work on this important topic. In my opinion, the manuscript in its current form is acceptable for publication, and I do not have further comments. Congratulations!